# Characterization of Fe(III) Adsorption onto Zeolite and Bentonite

**DOI:** 10.3390/ijerph17165718

**Published:** 2020-08-07

**Authors:** Tomáš Bakalár, Mária Kaňuchová, Anna Girová, Henrieta Pavolová, Rudolf Hromada, Zuzana Hajduová

**Affiliations:** 1Faculty of Mining, Ecology, Process Control and Geotechnologies, Technical University of Košice, 04200 Košice, Slovakia; tomas.bakalar@tuke.sk (T.B.); maria.kanuchova@tuke.sk (M.K.); anna.girova@tuke.sk (A.G.); henrieta.pavolova@tuke.sk (H.P.); 2Institute of Hygiene of Animals and Environment, University of Veterinary Medicine and Pharmacy in Košice, 041 81 Košice, Slovakia; rudolf.hromada@uvlf.sk; 3Faculty of Business Management, University of Economics in Bratislava, 852 35 Bratislava, Slovakia

**Keywords:** zeolite, bentonite, adsorption, Fe (III) removal, surface layer

## Abstract

In this study, the adsorption of Fe(III) from aqueous solution on zeolite and bentonite was investigated by combining batch adsorption technique, Atomic adsorption spectroscopy, X-ray diffraction, and X-ray photoelectron spectroscopy analyses. Although iron is commonly found in water and is an essential bioelement, many industrial processes require efficient removal of iron from water. Two types of zeolite and two types of bentonite were used. The results showed that the maximum adsorption capacities for removal of Fe (III) by Zeolite Micro 20, Zeolite Micro 50, blue bentonite, and brown bentonite were 10.19, 9.73, 11.64, and 16.65 mg.g^−1^, respectively. Based on the X-ray photoelectron spectroscopy (XPS) and X-ray fluorescence (XRF) analyses of the raw samples and the solid residues after sorption at low and high initial Fe concentrations, the Fe content is different in the surface layer and in the bulk of the material. In the case of lower initial Fe concentration (200 mg.dm^−3^), more than 95% of Fe is adsorbed in the surface layer. In the case of higher initial Fe concentration (4000 mg.dm^−3^), only about 45% and 61% of Fe is adsorbent in the surface layer of zeolite and bentonite, respectively; the rest is adsorbed in deeper layers.

## 1. Introduction

Iron [Fe(III)] is commonly found in almost all types of water, including surface and drinking water, but in some types of water, e.g., water used in heating systems and water-tube boilers, it is unfavorable. Iron is also a biogenic element. Its removal has been widely studied due to precipitation in water distribution systems [1,2]. Carbon is often used for removal of iron whether in form of activated carbon [3], graphene oxide [4] or biochar [5]. The most advantageous material is natural material, e.g., zeolite, bentonite, etc. from local sources that supports local economy and helps to lower carbon footprint. Zeolites, bentonites, and other natural material are widely used for removal of not only iron but also other heavy metals and inorganic pollutants. Zeolites and bentonites are adsorbents with a high uptake of Fe(III) due to high ion exchange capacity, high affinity to metal cations, and stability under different physical and chemical conditions [6,7]. Zeolites are often used in water treatment for drinking water purposes. Both zeolites and bentonites are also used in wastewater treatment [8,9,10]. A study [8] of Fe(III) removal by zeolite in its raw form and treated with Na_2_S_2_O_8_ and HNO_3_ at 20 °C and 70 °C reported a maximum sorption capacity of about 100 mg.g^−1^ except for treatment with HNO3 at 70 °C with a maximum sorption capacity of only 45 mg.g^-1^. Another study of Fe(III) adsorption [9] reported a sorption capacity of 28.9 and 30.0 mg.g^−1^ of Fe (III) for bentonite and H_2_SO_4_ activated bentonite. The influence of temperature on the sorption capacity is also significant as the maximum sorption capacity increases with increasing temperature [10] and this influence also depends on the nature of the process if it is endothermic or exothermic [11]. These and many other papers [12,13,14,15,16] only studied the sorption equilibrium and kinetics and, in some cases, influence of additives, in aqueous phase. If the solid phase was studied, only the material before the adsorption was characterized.

Removal of trivalent ions was studied by several papers [17,18,19,20,21]. These papers not only studied the sorption equilibrium in aqueous phase but also the solid surface before and after the adsorption. Ion exchange of Cr(III) on the surface of apple wood biochar was found when removing Cr(VI) from aqueous solution. Cr(VI) reduced to Cr(III) on the surface and biochar proved effective for Cr(VI) removal [17]. As (III) was successfully removed from aqueous solution by Fe-based backwashing sludge with a maximum sorption capacity of 59.7 mg.g^−1^ stating that the oxidation state of As(III) has not changed with the occurrence of the ion exchange between sulfate and arsenite [18]. Another effective As(III) adsorbent was Fe-modified W zeolite which proved a sorption capacity of 0.315 mg.g^−1^ and the adsorption took place by chemical bonding [19]. Eu(III) was adsorbed on maghemite with a maximum sorption capacity of 33.47 mg.g^−1^ and for the adsorption the oxygen-containing functional groups were responsible [20]. In other study, Eu(III) adsorption from aqueous solutions by biological calcium carbonate functionalized by mussel-inspired polydopamine with the maximum adsorption capacity determined to be 151.52 mg.g^−1^ was studied and the adsorption mechanism was proposed as the coordination of Eu(III) with the abundant catechol and amine/imine groups on the polydopamine coating [21].

For the reasons mentioned above, this study concentrates not only on the study of adsorption process in the aqueous phase but also in the solid phase, especially in the surface layer. The adsorption of Fe(III) ions onto natural zeolites and bentonites from local sources in Slovakia and Czech Republic was studied in batch experiments. The solutions were analyzed for the content of Fe and evaluated using adsorption isotherms. In addition, from each series, the raw material and two samples of solid residue after adsorption, at a low and high initial Fe(III) concentration were taken and analyzed for elemental composition by XPS and XRF to get a view on not only the sorption capacity but also the sorption process and the exchange of cations, while concentrating on crude, chemically not modified adsorbents.

## 2. Materials and Methods

Zeolites were provided by Zeocem, a.s. (Bystré, Slovakia). The zeolites were not modified. Bentonites were provided by KERAMOST, a.s. (Most, Czech Republic). The bentonites were crushed in ceramic mortar and sieved with 0.25 mm sieve (FRITSCH GmbH - Milling and Sizing, Weimar, Germany). The material from below the sieve was used for further analyses. In the study, two types of zeolite and two types of bentonite were used, namely Zeolite Micro 20 (Z-M20), Zeolite Micro 50 (Z-M50), blue bentonite (B-BL) and brown bentonite (B-BR).

The samples of bentonites and zeolites were analyzed for particle size distribution by a particle sizer Analysette 22 (Fritsch, Germany). The Sauter mean diameter of adsorbent particles, defined by equation
(1)$$d_{32}=\frac{\sum_{i=1}^{m}n_{i}d_{i}{}^{3}}{\sum_{i=1}^{m}n_{i}d_{i}{}^{2\;}},$$

*n_i_* is the mass percentage of *i*-th fraction (%), *d_i_* is the mean particle size of *i*-th fraction (m), can be used as mean particle size.

The initial Fe solution was prepared with analytical grade Fe_2_(SO_4_)_3_.9H_2_O provided by ITES Vranov, a.s. (Vranov, Slovakia). Water for the preparation of solutions was tap water purified by a five-stage reverse osmosis system (Aqualive, s.r.o., Košice, Slovakia). The equilibrium experiments were carried out with a series of PET flasks containing 0.1 dm^3^ (*V*) of metal ion solution of different initial concentrations (*C*_0_ = 1 to 4000 mg.dm^−3^) prepared from iron sulphate and a fixed dosage of sorbent (*C_a_* = 1 g.dm^−3^) which were agitated for 2 h in a rotary shaker at 3.33 s^−1^, with a temperature control at 25 °C, what was sufficient for the metal ions adsorption to reach an equilibrium, based on previous studies [6,8,10,11,12,13,15,16]. The initial pH of the solution was not adjusted. After equilibration and sedimentation of suspensions the samples were filtered through a micro filter of pore size 0.8 μm and the filtrates were analyzed for metal content by Atomic absorption spectroscopy (AAS). The solid samples [two from each series—at low (further on referred to as L, about 200 mg.dm^−3^) and high (further on referred to as H, about 4000 mg.dm^−3^) initial concentrations and the raw samples (further on referred to as R)] were analyzed by XPS and XRF. The amounts of metal adsorbed *q_e_* (mg.g^−1^) in each flask were determined from the difference between the initial metal concentration *C*_0_ (mg.dm^-3^) and metal concentration at equilibrium *C_e_* (mg.dm^−3^) in the solution and was calculated according to the following equation:(2)$$q_{e}=\frac{\left(C_{0}-C_{e}\right)V}{m_{a}},$$

*m_a_* (g) is the weight of adsorbent, *V* (L) is the volume of solution.

Adsorption isotherms are a critical piece of information on optimization of the use of adsorbents. Isotherm models provide an adequate description of Fe(III) adsorption equilibria on zeolites and bentonites. The following isotherms were used:
*Freundlich* [22]:(3)$$q_{e}=K_{f}C_{e}{}^{\frac{1}{n}},$$*K_f_* (mg^1−*n*^.L*^n^*.g^−1^) is adsorption capacity, *n* (1) is a constant related to the intensity of the adsorption; the isotherm represents sorption taking place on a heterogeneous surface with interaction between the adsorbed molecules [23];*Langmuir* [24]:(4)$$q_{e}=\frac{q_{m}.a_{L}.C_{e}}{1+a_{L}C_{e}},$$*q_m_* (mg.g^−1^) is maximum sorption capacity, *a_L_* (dm^3^.mg^−1^) is adsorption energy; the isotherm represents sorption taking place on a homogenous surface within the adsorbent [25],*Redlich-Peterson* [26]:(5)$$q_{e}=\frac{K_{R}.C_{e}}{1+a_{L}C_{e}^{\beta \;}},$$*K_R_* (dm^3^.g^−1^) and *a_R_* (dm^3^*^β^*.g^−^*^β^*) are constants, *β* (1) is exponent; the isotherm is used as compromise between the Langmuir and Freundlich systems [23].

Flame atomic absorption spectrometry (AAS) performed using iCE 3300 (ThermoFisher Scientific, Grand Island, NY, USA) was used to determine the concentration of Fe in the solutions within 24 h from the end of the adsorption process [air-acetylene oxidizing flame, lamp current 10 mA, wavelength 248.3 nm, standards prepared from 1 mg.cm^−3^ of Fe^3+^ in 2% HNO_3_ (20 °C) obtained from Thermo Fisher Scientific Inc., (Geel, Belgium) by dilution in distilled water]. XPS analysis of solid samples was performed using PHOIBOS 100 SCD (SPECS Surface Nano Analysis GmbH, Berlin, Germany) model equipped with a non-monochromatic X-ray source measured at 70 eV transition energy and high-resolution spectra at 50 eV, at room temperature. All spectra were acquired at a basic pressure of 2.10^−9^ mbar with AlKα excitation at 10 kV (100 W). The energy scale has been calibrated by normalizing the *C 1s* line of adsorbed adventitious hydrocarbons to 285.0 eV. The data were analyzed by SpecsLab2 CasaXPS software (Casa Software Ltd., Teignmouth, United Kingdom). The basic chemical composition of tested samples was investigated by XRF performed using SPECTRO iQ II (SPECTRO Analytical Instruments GmbH, Kleve, Germany) with SDD silicon drift detector with resolution of 145 eV at 10,000 pulses. Both XPS and XRF analyses are non-invasive techniques analyzing the composition of material. The difference between these two techniques is in the depth of scanning as XPS has a depth resolution in first 10 nm, whilst XRF enables a bulk analysis and has depth sensitivity [27,28,29].

All the experiments were performed in triples and the result was taken as the average value of each experiment.

## 3. Results

The Sauter mean diameter (*d_32_*), which anticipates spherical shape of particles, is different from the arithmetical mean diameter of adsorbent particles, which is a simple mean regardless of shape. The surface area of particles is similar. The basic physical properties of the used materials are presented in Table 1.

The basic chemical composition of raw zeolites and bentonites and the adsorbents after adsorption at low and high initial Fe(III) concentrations are presented in Table 2.

The particle size distributions are presented in Figure 1.

Though the particle size range of adsorbents is different, the particle size distribution is similar for zeolites and bentonites. Comparing the particle size range, *d_32_*, *d_50_* and the distribution, it can be stated that the Sauter mean particle size of zeolites is different with about 20 µm and 50 µm for Z-M20 and Z-M50, respectively. The arithmetic mean particle size of zeolites is different from the Sauter mean particle size but if comparing Z-M20 and Z-M50 the difference is not so significant with about 3.5 µm and 9.5 µm, respectively. Comparing the particle size distribution of Z-M20 and Z-M50, both have normal (Gaussian) distributions, i.e., the distribution is similar though the particle size is different. Comparing the bentonites properties, they have the same particle size ranges, the Sauter mean diameter is similar with 199 µm and 180 µm as well as the arithmetic mean particle size is similar with 19 µm and 13 µm for B-BL and B-BR, respectively. The particle size distributions of bentonites are not Gaussian distributions. Thus, the influence of the particle size and its distribution may influence the adsorption process and the amount of adsorbed Fe(III).

In the adsorption process, the maximum adsorption capacities for removal of Fe(III) by Z-M20, Z-M50, B-BL, and B-BR were 10.19 ± 0.40, 9.73 ± 0.39, 11.64 ± 0.50, and 16.65 ± 0.63 mg.g^−1^, respectively. Maximum adsorption capacities, expressed in molar quantities, were 0.18 ± 0.01, 0.17 ± 0.01, 0.21 ± 0.01, and 0.30 ± 0.01 mmol.g^−1^, respectively. The equilibrium data and the fitted data of Fe(III) adsorption by Freundlich, Langmuir, and Redlich Peterson isotherms are presented in Figure 2.

The coefficients of determination (R^2^) are in the range of 0.78 to 0.92, 0.57 to 0.91, 0.52 to 0.92, and 0.83 to 0.99 for Z-M20, Z-M50, B-BL, and B-BR, respectively. Based on the regression analysis and fit in the graphics, the experimental data are fit by Lagmuir isotherm the most accurately for adsorption onto Z-M50 and B-BL, assuming that the Fe(III) ions are adsorbed on a fixed number of sites, each site occupied by one adsorbed ion, all the sites are energetically equivalent, the adsorption is monolayer, and the adsorbed molecules do not interact [30] (Table 3). The most accurate fit is by Redlich-Peterson isotherm for adsorption onto Z-M20 and B-BR, which combines elements from both Langmuir and Freundlich equations; therefore, the mechanism of adsorption is a mix and does not follow ideal monolayer adsorption and is applicable in either homogenous or heterogeneous systems because of its versatility [31].

The XPS scan spectra are shown in Figure 3. In the XPS spectra of original Z-M20, Z-M50, B-BL, and B-BR, the highest peak represents the typical dominance of *O* 1s—oxygen that can be found throughout the zeolite and bentonite structures together with *Si* 2p—silicon and *Al* 2p—aluminium. The third highest concentration has *C* 1s—carbon. In bentonites, there is also a natural content of *Fe* 2p—iron with a higher concentration than *Ca* 2p—calcium and *Se* 3d—selenium. The Z-M20 and Z-M50 also contain *K* 2p—potassium but the content of *K* 2p—potassium, *Ca* 2p—calcium and *Se* 3d is low. In zeolites, iron is naturally in low concentrations, it can be considered as trace element.

The difference between Z-M20 and Z-M50, and B-BL and B-BR is only in the concentration of the elements.

The highest peaks in XPS analyses detect relevant elements in bentonite and zeolite surface and the structure presenting the highest concentrations of O, Si and Al. The same is confirmed by the XRF analyses with the aluminosilicate structure Si/Al (weight rate) = 6.55 and 3.24 [Si/Al (molar rate) = 6.29 and 3.11] on average for zeolite and bentonite, respectively. This ratio is constant independently from the amount of Fe adsorbed. On average the content of Si is 24.50%, 25.29%, 19.24% and 19.13% and Al is 4.29%, 3.56%, 5.97% and 5.99% for Z-M20, Z-M50, B-BL and B-BR, respectively. The content of Ca in pure samples is the highest except for case of B-BR which corresponds to the content of Fe. The content of Se fluctuates in both the surface layer and the bulk. K is only present in zeolites and its content is the highest in the pure samples. The content of Fe in the solid phase, based on the XPS analysis (Figure 3), is increasing with the increasing initial Fe concentration in the solution. The differences expressed in per cents of low and high initial Fe concentrations compared to the raw adsorbent are as follows:Z-M20: 15.16% and 9.26%, in this only case the percentage of Fe in the surface layer is lower in the case of higher initial Fe concentration,Z-M50: 62.97% and 200.00%,B-BL: 14.00% and 37.20%, andB-BR: 7.80% and 8.10%.

These results, though seeming confusing, are in line with the findings of XRF and the adsorption equilibrium experiments, as presented in the following.

Based on the basic difference in the theory of spectroscopy for XPS and XRF, as mentioned in the Experimental section, the Fe content is different in the surface layer and in the bulk of the material (Table 4).

Fe in the surface layer fluctuates but, in the bulk, it increases depending on the amount of adsorbed Fe which may be caused by the way of binding and the depth of adsorption layer. In the case of lower initial Fe concentration, more than 95% of Fe is adsorbed in the surface layer. In the case of higher initial Fe concentration, only about 45% and 61% of Fe is adsorbed in the surface layer of zeolite and bentonite, respectively; the rest is adsorbed in the deeper layers. It may also be caused by the way of sorption when for the lower initial Fe concentration only ion exchange occurs in the surface layer while for the higher Fe concentration not only ion exchange in the deeper layers may occur, but also some capture of physical or chemical origin. At in-between concentrations (initial Fe concentrations of 200–4000 mg.dm^−3^), a successive decrease of Fe concentration in the surface layer and an increase in the bulk can be expected meaning possible change of the adsorption type from ion exchange to physical and/or chemical sorption with increasing initial Fe concentration.

From the solution, more than 95% and more than 99% of Fe is removed by sorption on zeolite and bentonite, respectively, in the case of higher initial Fe concentration. Only about 25% and 35% of Fe is removed by sorption on zeolite and bentonite, respectively, in the case of lower initial Fe concentration. This fact corresponds to the theory of adsorption as in the case of the higher initial Fe concentration, the maximum sorption capacity has been reached, which cannot be said about the lower initial Fe concentration.

The Fe:Ca ratio (Table 5) increases with the increasing initial Fe concentration, which means that with a rising amount of adsorbed Fe, the amount of desorbed Ca both in the surface layer and in the bulk grows. Only for B-BR, the ratio decreased in case of lower initial Fe concentration in the surface layer.

The change in Fe:Se ratio (Table 6) depends on the layer; whilst in the surface layer the ratio decreases, in the bulk, it increases. This implies that Se is dominantly decreasing in the inner layers.

The Fe:K ratio (Table 7) increases with the increasing initial Fe concentration, which means that with a growing amount of adsorbed Fe, the amount of desorbed K both in the surface layer and in the bulk grows as well.

## 4. Discussion

The above mentioned implies that the Fe(III) ions occupy the vacancies after Ca(II) and K(I) dissolved in solution. The situation is similar to Bi(III) ion-exchange for Ca^2+^, Mg^2+^ and Fe^2+^, the impurity atoms on rutile surface, dissolved in acidic solution [32]. Ion exchange between Cr(III) and K^+^, Ca^2+^ and Na^+^ in the aqueous solution after Cr(VI) removal from aqueous solution using apple wood biochar was reported [17]. The release of Se(metal) is disputable.

Based on a comparison of adsorption capacities (Table 8), the maximum adsorption capacities of studied zeolites and bentonites are lower than the ones of compared adsorbents. The sorption capacity might be increased by their modification by some of the following procedures:Prior to the coating treatment, the zeolite or bentonite is treated by suspending in NaCl solution for a period of 24 h. The suspension is filtered and washed with deionized water. The resulting suspension is dried in oven at 100 °C [33]. Manganese oxide coated zeolite and bentonite are prepared utilizing a reductive procedure [34] modified to precipitate colloids of manganese oxides onto zeolite and bentonite surfaces. Manganese oxide is precipitated in aqueous solution by the reaction:2KMnO_4_ + 8HCl = 2MnO_2_ + 2KCl + 3Cl_2_ + 4H_2_ODried zeolites and bentonites samples are poured over a heated solution at 90 °C, containing potassium permanganate placed in a beaker, followed by dropwise addition of hydrochloric acid. After stirring for 1 h, the suspension is filtered, washed several times using distilled water, and dried in an oven at 100 °C [35]. MnO_2_ coating increases not only the Mn(II), but also Fe(II) and Fe(III) removal.Prior to modification, the zeolite or bentonite is washed with deionized water and dried at room temperature. The modified zeolite or bentonite are obtained by pouring a mixture of MnCl_2_ and NaOH over the washed zeolite in a heat resistant dish and then heating the mixture in a furnace at 150 °C for approximately 5 h. Afterwards, the modified adsorbent is heated at 500 °C for 3 h, cooled at room temperature, and washed several times with distilled water [36].The modified zeolite or bentonite is prepared by mixing with Fe(III) solution. The mixture is shaken for 20 h at 25 °C before pH is measured and NaOH solution is added to raise the pH. This procedure is repeated every 2 h for a total of three times to bring the final solution pH to 9. The mixture is allowed to settle, and the supernatant removed, followed by washing the adsorbent with de-ionized water [37].The natural zeolite or bentonite is first treated with NaCl solution under reflux for 3 h. The treated solution is filtered, washed with distilled water, and dried at 60 °C for 24 h. Next, it is treated with FeCl_3_ solution under reflux for 5 h. The mixture is filtered, washed with distilled water, and dried at 80 °C. The procedure mentioned before is applied in a similar manner using MnCl_2_ and a mixture of FeCl_3_ and MnCl_2_ solutions [38].

Other methods may be used, or the above procedures may be modified as necessary to achieve maximum capacity of the modified zeolite and bentonite for adsorption of Fe(III).

This fact may be caused by different conditions of the adsorption, especially particle size distribution, pH, and temperature of the adsorption process, which significantly influences the maximum sorption capacity and the whole process. In this study, the conditions were not changed in order to minimize the economic side of the process, i.e., to use minimum of physical and chemical processes and chemical substances for pre-treatment of adsorbent or adsorbate, which is not the subject of this study.

Though the desorption process of Fe(III) from zeolites and bentonites was not part of this study, it could be realized with diethylenetriaminepentaacetic acid [39], 0.5 mol.l^−1^ HNO_3_ [40], Na_2_-EDTA [41], HCl and EDTA [42], thermally [43] etc. The desorption may be a part of further study.

## 5. Conclusions

The research of Fe(III) removal from water is important for adjustment of organoleptic properties of drinking water and for prevention of iron precipitation in water for industrial purposes. The use of natural material from local sources is a challenge; for this reason, natural zeolites and bentonites were used for Fe(III) adsorption. Both the aqueous and solid phases were studied using available techniques. The studied zeolites and bentonites are suitable for Fe(III) removal as their capacities are an average of 10 and 14 mg.g^-1^, respectively. The amount of Fe adsorbed in the surface layer and in the bulk is different depending on the initial Fe concentration, namely only 45% and 61% of Fe is adsorbed in the surface layer of zeolites and bentonites, respectively, in the case of high initial Fe concentration; and more than 95% of Fe is adsorbed in the surface layer of both zeolites and bentonites in the case of low initial Fe concentration.

In conclusion, the studied zeolites and bentonites are viable and can be simply used for the removal of iron from water contaminated with even higher concentration of Fe. Nevertheless, additional studies should be conducted to further characterize the adsorbents, especially reusability and adsorption capacity after several cycles.

## Figures and Tables

**Figure 1 ijerph-17-05718-f001:**
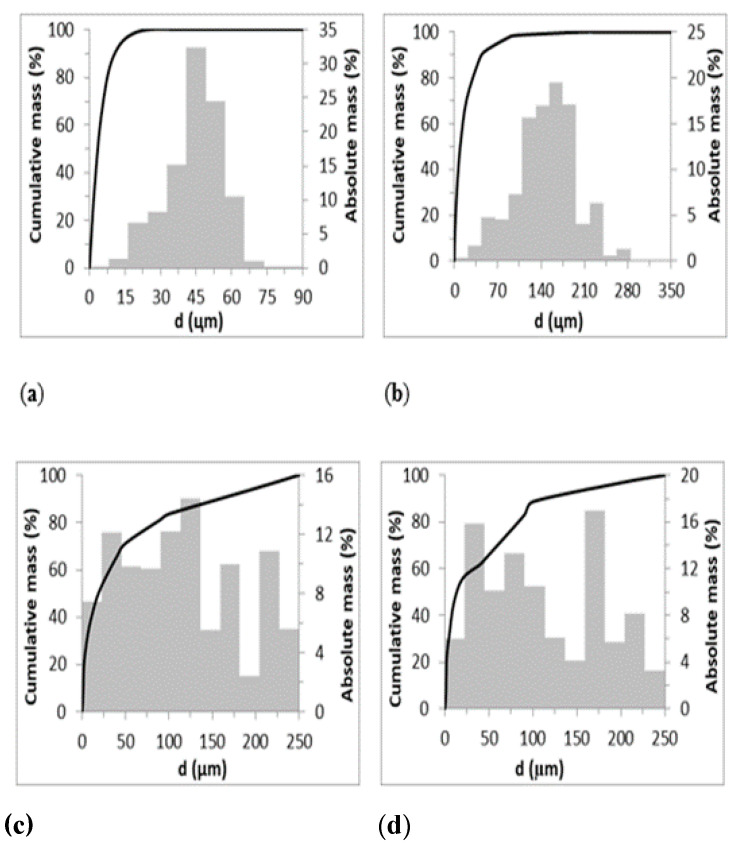
Particle size distribution of (**a**) Z-M20, (**b**) Z-M50, (**c**) B-BL, (**d**) B-BR.

**Figure 2 ijerph-17-05718-f002:**
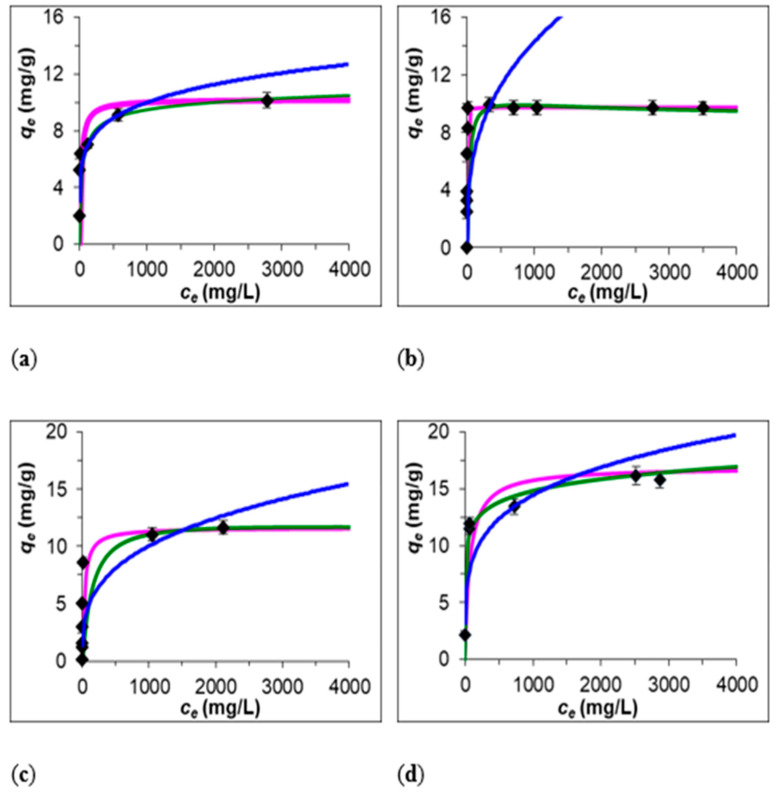
Experimental data of Fe(III) adsorption on (**a**) Z-M20, (**b**) Z-M50, (**c**) B-BL, (**d**) B-BR. Source: Own processing. Note: Black dots—experimental data, blue line—Freundlich isotherm, pink line—Langmuir isotherm, green line—Redlich-Peterson isotherm.

**Figure 3 ijerph-17-05718-f003:**
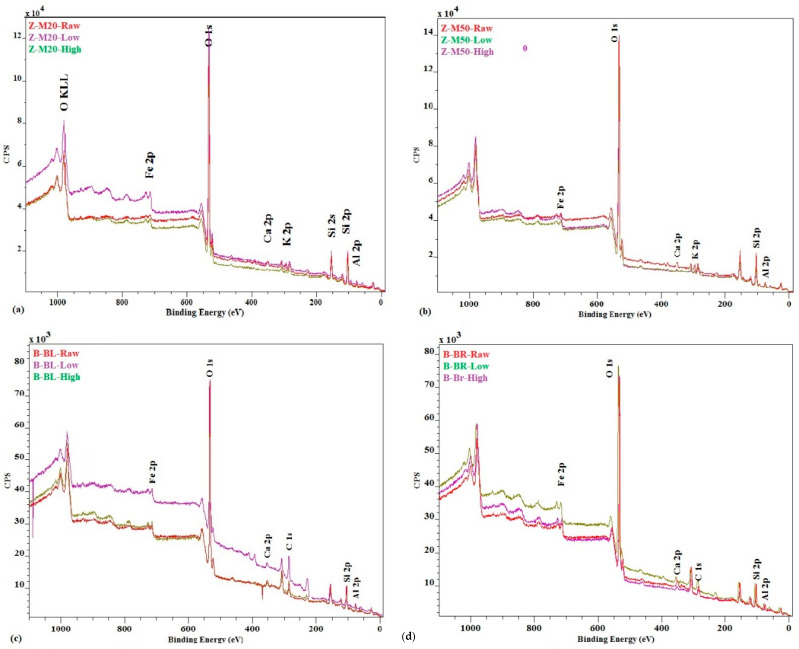
XPS analyses of (**a**) Z-M20, (**b**) Z-M50, (**c**) B-BL, (**d**) B-BR.

**Table 1 ijerph-17-05718-t001:** The basic physical properties of zeolites and bentonites.

Parameter	Z-M20	Z-M50	B-BL	B-BR
Particle size (µm)	0-90	0–350	0–250	0–250
*d*_32_ (µm)	19.553	50.862	199	180
*d*_50_ (µm)	3.493	9.549	19	13
Surface area (m^2^.g^−1^)	25.8394	26.3208	21.8874	20.1231

Abbr.: Z-M20–Zeolite Micro 20; Z-M50–Zeolite Micro 50, B-BL–blue bentonite, B-BR–brown bentonite.

**Table 2 ijerph-17-05718-t002:** The chemical composition of raw and modified zeolites and bentonites.

Compound	Z-M20			Z-M50			B-BL			B-BR		
	Raw	Low	High	Raw	Low	High	Raw	Low	High	Raw	Low	High
SiO_2_ (%)	51.54	51.31	51.73	54.33	53.38	55.82	41.27	40.63	40.94	44.78	41.29	40.87
Al_2_O_3_ (%)	8.66	8.11	7.92	7.35	7.13	6.76	10.54	11.37	11.32	11.66	11.87	11.71
CaO (%)	1.79	0.26	0.45	1.26	0.15	0.44	1.97	1.51	1.02	2.53	1.69	1.38
K_2_O (%)	1.36	0.42	0.29	1.45	0.18	0.30	0.00	0.00	0.00	0.00	0.00	0.00
Fe_2_O_3_ (%)	0.83	0.99	2.27	2.27	0.08	1.47	2.92	3.26	5.22	2.17	2.48	4.81
FeO (%)	0.10	0.10	0.10	0.10	0.10	0.10	0.10	0.10	0.10	0.10	0.10	0.10

**Table 3 ijerph-17-05718-t003:** Adsorption isotherm parameters of Fe(III) on zeolites and bentonites.

Isotherm	Parameter	Z-M20	Z-M50	B-BL	B-BR
Freundlich	*K_f_*, mg^1−n^.dm^3n^.g^−1^	3.07	1.29	1.17	3.05
	*n*	5.85	2.87	3.21	4.44
	*R^2^*	0.78	0.57	0.52	0.83
Langmuir	*q_m_*, mg.g^−1^	10.19	9.72	11.64	16.86
	*a_L_*, dm^3^.mg^−1^	0.05	1.18	0.03	0.01
	*R^2^*	0.92	0.91	0.91	0.95
Redlich-Peterson	*K_R_*, dm^3^.g^−1^	0.31	0.32	0.08	2.50
	*b_R_*, dm^3^^β^.g^−^^β^	0.05	0.02	0.01	0.32
	*β*	0.94	1.04	1.04	0.91
	*R^2^*	0.92	0.82	0.87	0.99

**Table 4 ijerph-17-05718-t004:** A comparison of Fe content in mg/g ± standard deviation.

Method	Z-M20			Z-M50			B-BL			B-BR		
	Raw	Low	High	Raw	Low	High	Raw	Low	High	Raw	Low	High
XPS	7.02 ± 0.31	8.08 ± 0.31	7.67 ± 0.25	1.85 ± 0.05	3.02 ± 0.10	5.55 ± 0.21	15.00 ± 0.63	17.10± 0.65	20.58 ± 0.91	20.38 ± 0.97	21.97 ± 0.90	22.03 ± 0.94
XRF	7.11 ± 0.29	8.27 ± 0.29	17.19 ± 0.47	1.89 ± 0.06	3.06 ± 0.09	11.61 ± 0.52	15.90 ± 0.63	18.07 ± 0.74	34.31 ± 1.06	21.11 ± 0.96	23.47 ± 0.99	37.22 ± 1.09
Adsorbed		1.16	10.08		1.17	9.72		2.17	18.41		2.36	16.10
XPS + ads		8.18	17.10		3.02	11.57		17.17	33.41		22.74	36.48
XRF + ads		8.27	17.19		3.06	11.61		18.07	34.31		23.47	37.21

**Table 5 ijerph-17-05718-t005:** A comparison of Fe:Ca ratio.

Ratio, Method	Z-M20			Z-M50			B-BL			B-BR		
	Raw	Low	High	Raw	Low	High	Raw	Low	High	Raw	Low	High
Fe:Ca, XPS	0.52	2.06	1.95	0.21	0.70	1.23	0.84	1.27	1.95	1.48	1.26	2.43
Fe:Ca, XRF	0.56	4.45	5.39	0.21	2.81	3.69	0.88	1.49	3.47	1.50	2.17	5.08
ΔFe:Ca, XPS, %		298.64	276.31		237.75	490.02		51.04	86.74		−14.80	64.29
ΔFe:Ca, XRF, %		701.42	871.05		1238.92	1659.31		70.14	294.72		45.15	239.62

**Table 6 ijerph-17-05718-t006:** A comparison of Fe:Se ratio.

Ratio, Method	Z-M20			Z-M50			B-BL			B-BR		
	Raw	Low	High	Raw	Low	High	Raw	Low	High	Raw	Low	High
Fe:Se, XPS	3.21	0.79	2.22	1.01	0.42	0.75	3.31	3.62	2.42	3.89	2.28	2.12
Fe:Se, XRF	3.57	3.97	8.76	0.24	0.44	1.53	2.88	4.35	5.02	1.79	2.55	4.18
ΔFe:Se, XPS, %		−75.42	−30.84		−58.00	−25.51		9.41	−26.97		−41.34	−45.48
ΔFe:Se, XRF, %		11.17	145.27		82.63	538.68		51.17	74.46		42.69	133.9

**Table 7 ijerph-17-05718-t007:** A comparison of Fe:K ratio.

Ratio, Method	Z-M20			Z-M50		
	Raw	Low	High	Raw	Low	High
Fe:K, XPS	0.57	1.07	1.12	0.16	0.51	1.56
Fe:K, XRF	2.38	7.25	0.16	0.16	2.04	4.64
ΔFe:K, XPS, %		87.19	94.95		227.88	898.60
ΔFe:K, XRF, %		278.50	1055.91		1196.55	2853.99

**Table 8 ijerph-17-05718-t008:** A comparison of sorption capacities of different Fe adsorbents.

Adsorbent	*q_m_* mg.g^−1^	Temperature °C	Initial pH	Source
Zeolite—M20	10.19	25	*	this study
Zeolite—M50	9.73	25	*	this study
Bentonite—BL	11.64	25	*	this study
Bentonite—BR	16.86	25	*	this study
Zeolite	98.00	room	3.0	[8]
Bentonite	28.90	30	3.0	[9]
H_2_SO_4_ activated bentonite	30.00	30	3.0	[9]
Egyptian Bentonite	52.63	20	4.0	[10]
Egyptian Bentonite	56.18	40	4.0	[10]
Egyptian Bentonite	58.48	50	4.0	[10]
Egyptian Bentonite	63.69	60	4.0	[10]
Sawdust modified with diethylenetriamine	200.00	room	3.0	[12]
Elderberry pomace	33.25	23	3.4	[13]
Chitosan/Fe_3_O_4_/graphene oxide nanocomposite	6.50	room	2.5	[15]
Y zeolite	31.45	25	6.5	[16]

* No pH adjustment.

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
