# Peer review of "Characterization of Fe(III) Adsorption onto Zeolite and Bentonite"

_ijerph, 2020, doi:10.3390/ijerph17165718_

Round 1

Reviewer 1 Report

In my opinion, they are only laboratory tests that do not have sufficient entity as a research device.

Author Response

Dear Reviewer,

thank you very much for your time and effort regarding our work.

We are very sorry that our post did not address you. In our opinion, this contribution is highly topical following Agenda 2030 with goal 6: Ensure availability and sustainable management of water and sanitation for all. Sustainable water resources are essential to human health, environmental sustainability and economic prosperity. This vital resource is under threat, making it crucial to address the challenges related to water, sanitation and water-related ecosystems. We think that the experimental work, for which we have the equipment at our university, can be the basis for the solution of water treatment and its reuse on a national as well as global scale.

It is on the basis of experiments that scientific conclusions and claims can be made.

Yours sincerely,

Authors

Reviewer 2 Report

The authors report the adsorption on Fe(III) on natural zeolites and bentonites, being their aim the characterization of the adsorbent after  iron removal and comparing it to the raw adsorbent. They use XPS and XRF techniques to identify the changes in the chemical composition of the adsorbents. However the manuscript requires major revisions that improve the quality and discussion of the results.

  • The chemical formula and units should be written using the proper subscript and super index in the abstract, introduction, and Table 2,
  • On page 2 lines 44-45 the authors mention “The influence of temperature on the sorption capacity is also significant as the maximum sorption capacity increases with increasing temperature” However the influence of the temperature on the sorption capacity also depend on the nature of the process if it is endothermic or exothermic, please check
  • On page 3, line 108 , The name of isotherm is not correct it should be Langmuir, please verify the number of the reference too, reference [23] is missing
  • There is confusion about the XRD and XRF techniques throughout the manuscript. The technique that is reported in the manuscript is XRF not XRD, the information that can be obtained from each technique is very different. It is important that the authors check and correct it. For instance:

Abstract:”… Atomic adsorption spectroscopy, X-ray  diffraction and X-ray photoelectron spectroscopy analyses…”.

“…  Based on the XPS and XRD analyses of the raw samples and the solid residues after sorption at low and high  initial Fe concentrations, the Fe content is different in the surface layer and in the bulk of the material”

Introduction: “…analyzed for elemental composition by XPS and XRD to get a view on not only the sorption capacity but also the sorption process and the exchange of cations

Page 7; Lines 177-179 The same is confirmed by the XRD analyses with the alumosilicate structure Si/Al (weight rate) = 6.55 and 3.24 [Si/Al (molar rate) = 6.29 and 3.11] on average for zeolite and bentonite, respectively.

  • On Table 1 the authors present the chemical composition of the zeolites and bentonites used, However the percentage for each oxide is reported as a range of percentage, why is not reported the specific percentage for each oxide for each material?, considering that the data obtained from XRF are accurate
  • The discussion of the particle size distribution (Fig 1) should be improved, there are significant differences in the distribution for each material
  • The XRD characterization of the bentonites and zeolites before and after adsorption process should be included on the manuscript, considering that the aim of this research is the characterization of the adsorbent before and after iron removal
  • The quality of the XPS spectra should be improved, it would be convenience that the Spectra for each material (raw and after iron removal at low and high concentrations) will be presented in only one graph, the changes in the peaks would be clearer. For XPS analysis is important that each peak in the spectra will be identified, besides indicate in the figure caption the meaning of R, L and H. Additionally in the discussion the authors should be include the BE corresponding to each detected element. The discussion should be improved, the XPS technique could provide additional information about the mechanism of iron adsorption on the bentonite and zeolite which would explain the difference in the adsorption capacity. A peaks deconvolution would be useful.
  • There are some English typo errors in the manuscript, please revise in detail

For instance:

Page 7, Lines 177-179: The same is confirmed by the XRD analyses with the alumosilicate structure Si/Al (weight rate) = 6.55 and 3.24 [Si/Al (molar rate) = 6.29  and 3.11] on average for zeolite and bentonite, respectively

Page 9, lines 228-229: Based on a comparison of adsorption capacities (Table 7.) the maximum adsorption capacities of studied zeolites and bentonites are lower than the ones of compered adsorbents.

Author Response

Dear Reviewer:

Thank you for your helpful comments concerning our manuscript. We sincerely thank the reviewers for their helpful comments, which have helped us clarify many points and improve the paper.

Based on the comments and suggestions, we have revised our paper carefully. In what follows, we describe in greater detail how the revised paper has addressed the reviewers’ comments. Our responses to the reviewers’ comments are described in detail below. We hope these revisions are adequate.

Thank you very much for your time and efforts concerning our paper. We look forward to hearing from you regarding our revised manuscript. We would be glad to respond to any further question and comment that you may have.

Yours sincerely,

Authors

The authors report the adsorption on Fe(III) on natural zeolites and bentonites, being their aim the characterization of the adsorbent after  iron removal and comparing it to the raw adsorbent. They use XPS and XRF techniques to identify the changes in the chemical composition of the adsorbents. However the manuscript requires major revisions that improve the quality and discussion of the results.

  • The chemical formula and units should be written using the proper subscript and super index in the abstract, introduction, and Table 2:

 All the formulas and units were corrected.

  • On page 2 lines 44-45 the authors mention “The influence of temperature on the sorption capacity is also significant as the maximum sorption capacity increases with increasing temperature” However the influence of the temperature on the sorption capacity also depend on the nature of the process if it is endothermic or exothermic, please check:

The following text was added: “and this influence also depends on the nature of the process if it is endothermic or exothermic [11]”.

  • On page 3, line 108 , The name of isotherm is not correct it should be Langmuir, please verify the number of the reference too, reference [23] is missing:

 The name of the isotherm was corrected to Langmuir, the numbers of references were changed, reference [23] is now reference [24]: Langmuir I. The constitution and fundamental properties of solids and liquids. Part I. Solids. J. Am. Chem. Soc., 38, 2221, 1916.

  • There is confusion about the XRD and XRF techniques throughout the manuscript. The technique that is reported in the manuscript is XRF not XRD, the information that can be obtained from each technique is very different. It is important that the authors check and correct it. For instance:

Thank you for the comment. The used method is XRF, and the abbreviation XRD was used by mistake.

Abstract:”… Atomic adsorption spectroscopy, X-ray  diffraction and X-ray photoelectron spectroscopy analyses…”.

“…  Based on the XPS and XRD analyses of the raw samples and the solid residues after sorption at low and high  initial Fe concentrations, the Fe content is different in the surface layer and in the bulk of the material”

Introduction: “…analyzed for elemental composition by XPS and XRD to get a view on not only the sorption capacity but also the sorption process and the exchange of cations

Page 7; Lines 177-179 The same is confirmed by the XRD analyses with the alumosilicate structure Si/Al (weight rate) = 6.55 and 3.24 [Si/Al (molar rate) = 6.29 and 3.11] on average for zeolite and bentonite, respectively.

  • On Table 1 the authors present the chemical composition of the zeolites and bentonites used, However the percentage for each oxide is reported as a range of percentage, why is not reported the specific percentage for each oxide for each material?, considering that the data obtained from XRF are accurate:

The data were taken for different samples of zeolites and bentonites, that were statistically evaluated that is why there was a range of percentage; the numbers were changed for the particular samples used at the experiment.

  • The discussion of the particle size distribution (Fig 1) should be improved, there are significant differences in the distribution for each material

Thank you for the comment, the following text was added: “Comparing the particle size range, d32, d50 and the distribution it can be stated that the Sauter mean particle size of zeolites is different with about 20 µm and 50 µm for Z-M20 and Z-M50, respectively. The arithmetic mean particle size of zeolites is different from the Sauter mean particle size but if comparing Z-M20 and Z-M50 the difference is not so significant with about 3.5 µm and 9.5 µm, respectively. Comparing the particle size distribution of Z-M20 and Z-M50, both have normal (Gaussian) distributions, i.e. the distribution is similar though the particle size is different. Comparing the bentonites properties they have the same particle size ranges, the Sauter mean diameter is similar with 199 µm and 180 µm as well as the arithmetic mean particle size is similar with 19 µm and 13 µm for B-BL and B-BR, respectively. The particle size distributions of bentonites are not Gaussian distributions. Thus the influence of the particle size and its distribution may influence the adsorption process and the amount of adsorbed Fe(III).”

  • The XRD characterization of the bentonites and zeolites before and after adsorption process should be included on the manuscript, considering that the aim of this research is the characterization of the adsorbent before and after iron removal

 The Table 1 was split into two tables, one of them presenting the physical properties and the other one the chemical composition.

  • The quality of the XPS spectra should be improved, it would be convenience that the Spectra for each material (raw and after iron removal at low and high concentrations) will be presented in only one graph, the changes in the peaks would be clearer. For XPS analysis is important that each peak in the spectra will be identified, besides indicate in the figure caption the meaning of R, L and H. Additionally in the discussion the authors should be include the BE corresponding to each detected element. The discussion should be improved, the XPS technique could provide additional information about the mechanism of iron adsorption on the bentonite and zeolite which would explain the difference in the adsorption capacity. A peaks deconvolution would be useful:

The figures were changed, the Spectra for each material is presented in only one graph, differentiated by colour and the following text was added “The content of Fe in the solid phase, based on the XPS analysis (Figure 3), is increasing with the increasing initial Fe concentration in the solution. The differences expressed in per cents of low and high initial Fe concentrations compared to the raw adsorbent are as follows:

  • Z-M20: 15.16% and 9.26%, in this only case the percentage of Fe in the surface layer is lower in the case of higher initial Fe concentration,
  • Z-M50: 62.97% and 200.00%,
  • B-BL: 14.00% and 37.20%, and
  • B-BR: 7.80% and 8.10%.

These results, though seeming confusing, are in line with the findings of XRF and the adsorption equilibrium experiments, as presented in the following.

  • There are some English typo errors in the manuscript, please revise in detail:

 The typos were revised.

For instance:

Page 7, Lines 177-179: The same is confirmed by the XRD analyses with the alumosilicate structure Si/Al (weight rate) = 6.55 and 3.24 [Si/Al (molar rate) = 6.29  and 3.11] on average for zeolite and bentonite, respectively

Page 9, lines 228-229: Based on a comparison of adsorption capacities (Table 7.) the maximum adsorption capacities of studied zeolites and bentonites are lower than the ones of compered adsorbents.

Reviewer 3 Report

1, Mark full name of "R, L H" in Table 3,4,5,6

2,  Discus on possibility of Zeolites and bentonites, How to desorb. 

3, Discus on how to improve its adsorption capacity becasue tested zeolites and bentonites capacity is low

Author Response

Dear Reviewer:

Thank you for your helpful comments concerning our manuscript. We sincerely thank the reviewers for their helpful comments, which have helped us clarify many points and improve the paper.

Based on the comments and suggestions, we have revised our paper carefully. In what follows, we describe in greater detail how the revised paper has addressed the reviewers’ comments. Our responses to the reviewers’ comments are described in detail below. We hope these revisions are adequate.

Thank you very much for your time and efforts concerning our paper. We look forward to hearing from you regarding our revised manuscript. We would be glad to respond to any further question and comment that you may have.

Yours sincerely,

Authors

1, Mark full name of "R, L H" in Table 3,4,5,6.

      The full names were added R = Raw, L = Low, H =  High.

2,  Discus on possibility of Zeolites and bentonites, How to desorb.

     The following text was added: “Though the desorption process of Fe(III) from zeolites and bentonites was not part of this study, it could be realized with diethylenetriaminepentaacetic acid [40], 0.5 mol.l-1 HNO3 [41], Na2-EDTA [42], HCl and EDTA [43], thermally [44], etc. The desorption may be a part of further study.”

3, Discus on how to improve its adsorption capacity becasue tested zeolites and bentonites capacity is low.

 The following text was added: “The sorption capacity might be increased by their modification by some of the following procedures.

  • Prior to the coating treatment, the zeolite or bentonite will be treated by suspending in NaCl solution for a period of 24 h. The suspension will be filtered and washed with deionized water. The resulting suspension will be dried in oven at 100 °C [34]. Manganese oxide coated zeolite and bentonite will be prepared utilizing a reductive procedure [35] modified to precipitate colloids of manganese oxides onto zeolite and bentonite surfaces. Manganese oxide will be precipitated in aqueous solution by the reaction:
    2KMnO4 + 8HCl = 2MnO2 + 2KCl + 3Cl2 + 4H2O
    Dried zeolites and bentonites samples will be poured over a heated solution at 90 °C, containing potassium permanganate placed in a beaker, followed by dropwise addition of hydrochloric acid. After stirring for 1 h, the suspension will be filtered, washed several times using distilled water, dried in oven at 100 °C [36]. MnO2 coating increases not only the Mn(II) but also the Fe(II) and Fe(III) removal.
  • Prior to modification, the zeolite or bentonite will be washed with deionized water and dried at room temperature. The modified zeolite or bentonite will be obtained by pouring a mixture of MnCl2 and NaOH over the washed zeolite in a heat resistant dish and then heating the mixture in a furnace at 150 °C for approximately 5 h. Afterwards, the modified adsorbent will be heated at 500 °C for 3 h, cooled at room temperature and washed several times with distilled water [37].
  • The modified zeolite or bentonite will be prepared by mixing with Fe(III) solution. The mixture will be shaken for 20 h at 25 °C before pH is measured and NaOH solution will be added to raise the pH. This procedure will be repeated every 2 h for a total of three times to bring the final solution pH to 9. The mixture will be allowed to settle, and the supernatant removed, followed by washing the adsorbent with de-ionized water [38].
  • The natural zeolite or bentonite will be first treated with NaCl solution under reflux for 3 h. The treated solution will be filtered, washed with distilled water, and dried at 60 °C for 24 h. Next it will be treated with FeCl3 solution under reflux for 5 h. The mixture will be filtered, washed with distilled water, and dried at 80 °C. The procedure mentioned before will be applied in a similar manner using MnCl2 and a mixture of FeCl3 and MnCl2 solutions [39].

Other methods may be used or the above procedures may be modified as necessary to achieve the maximum capacity of modified zeolite and bentonite for adsorption of Fe(III).”

Round 2

Reviewer 1 Report

Dear authors, thank you for your response. I think the effort is good but the result does not have the level to be published.

Author Response

Thank you for your opinion concerning our manuscript. We must say that your comments have not been specifically defined, which we are very sorry about, because our goal is to satisfy you as a reviewer.

We sincerely thank the reviewers for their helpful comments, which have helped us clarify many points and improve the paper. Based on the comments and suggestions, we have revised our paper carefully.

Thank you very much for your time and efforts concerning our paper. We look forward to hearing from you regarding our revised manuscript. We would be glad to respond to any further question and comment that you may have.

Thank you,

Sincerely,

 Authors

Reviewer 2 Report

The manuscript has been substantially improved. The results are clearer and discussed in detail.  I just suggest an additional revision for typo, for instance:

Page 7 lines 256-257 : “….The same is confirmed by the XRF 256 analyses with the alumosilicate structure Si/Al (weight rate) = 6.55 and 3.24 [Si/Al (molar rate) = 6.29…..”

Author Response

Dear Reviewer:
Thank you for your helpful comments concerning our manuscript.
Thank you very much for your time and efforts concerning our paper. We look forward to hearing from you regarding our revised manuscript. We would be glad to respond to any further question and comment that you may have.
Yours sincerely,
Authors

Specific comments
Page 7 lines 256-257 : “….The same is confirmed by the XRF 256 analyses with the alumosilicate structure Si/Al (weight rate) = 6.55 and 3.24 [Si/Al (molar rate) = 6.29…..”

Thank you for this helpful comment, we kindly ask you, if you can read this section, if this section meets scientific requirements now.
